# Structure Consistent Gaussian Splatting with Matching Prior for Few-shot Novel View Synthesis

**Rui Peng[1,2]  Wangze Xu[1]  Luyang Tang[1,2]  Liwei Liao[1]  Jianbo Jiao[3]  Ronggang Wang[✉,1,2]**

[1]Guangdong Provincial Key Laboratory of Ultra High Definition Immersive Media
Technology, Peking University Shenzhen Graduate School
[2]Peng Cheng Laboratory  [3]University of Birmingham
ruipeng@stu.pku.edu.cn   rgwang@pkusz.edu.cn

## Abstract

Despite the substantial progress of novel view synthesis, existing methods, either based on the Neural Radiance Fields (NeRF) or more recently 3D Gaussian Splatting (3DGS), suffer significant degradation when the input becomes sparse. Numerous efforts have been introduced to alleviate this problem, but they still struggle to synthesize satisfactory results efficiently, especially in the large scene. In this paper, we propose *SCGaussian*, a Structure Consistent Gaussian Splatting method using matching priors to learn 3D consistent scene structure. Considering the high interdependence of Gaussian attributes, we optimize the scene structure in two folds: rendering geometry and, more importantly, the position of Gaussian primitives, which is hard to be directly constrained in the vanilla 3DGS due to the non-structure property. To achieve this, we present a hybrid Gaussian representation. Besides the ordinary non-structure Gaussian primitives, our model also consists of ray-based Gaussian primitives that are bound to matching rays and whose optimization of their positions is restricted along the ray. Thus, we can utilize the matching correspondence to directly enforce the position of these Gaussian primitives to converge to the surface points where rays intersect. Extensive experiments on forward-facing, surrounding, and complex large scenes show the effectiveness of our approach with state-of-the-art performance and high efficiency. Code is available at https://github.com/prstrive/SCGaussian.

## 1 Introduction

Few-shot novel view synthesis (NVS) aims to reconstruct the scene given only a sparse collection of views, which has always been a cornerstone and challenging task in computer vision. Neural radiance field (NeRF) [29], emerged as an excelled 3D representation, has shown great success in rendering photo-realistic novel views. However, such impressive results require an expensive and time-consuming collection of dense images which impedes many practical applications, *e.g.*, the input is typically much sparser in autonomous driving, robotics, and virtual reality. Although many attempts have been proposed to solve this challenging few-shot rendering problem from the aspect of pre-training [6, 52, 60, 17], regularization terms [20, 32, 58, 43], external priors [8, 39, 51, 46, 15], *etc.*, these NeRF-based methods still suffer from low rendering speed and high computational cost, *i.e.*, each scene requires hours or even days of training time.

Recently, an efficient representation 3D Gaussian Splatting (3DGS) [19] is proposed to leverage a set of Gaussian primitives (initialized from the Structure-from-motion (SFM) [41, 42] points) along with some attributes to explicitly model the 3D scene. Through replacing the cumbersome volume rendering in NeRF methods with the efficient differentiable splatting, which directly projects the Gaussian primitives onto the 2D image plane, 3DGS has expressed remarkable improvement in both

38th Conference on Neural Information Processing Systems (NeurIPS 2024).

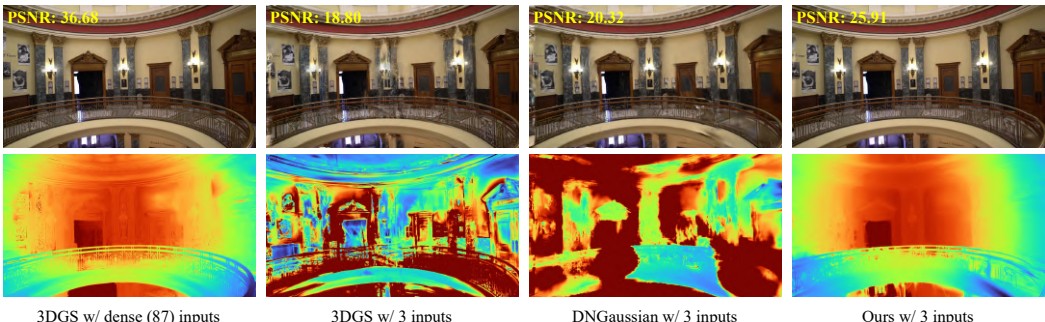

| 3DGS w/ dense (87) inputs | 3DGS w/ 3 inputs | DNGaussian w/ 3 inputs | Ours w/ 3 inputs |

Figure 1: **Comparisons in view synthesis and geometry rendering.** 3DGS [19] can synthesize high-quality novel views and plausible geometry with excessive inputs, but suffers from significant degradation in the sparse scenario. Even using the monocular depth prior, DNGaussian [23] still struggles to generate accurate geometry and novel views. In contrast, our method can learn the more consistent scene structure and render the more realistic images.

rendering quality and speed, *i.e.*, high-resolution images can be rendered in real-time. Even with this unprecedented performance, 3DGS still relies on dense image captures and faces the same problem of novel view degeneration with NeRF methods, when only a few inputs are available.

In this paper, we aim to address this issue by establishing a few-shot 3DGS model with a consistent structure to pursue high-quality and efficient novel view synthesis. Compared with the dense counterpart, this few-shot system introduces more challenging problems, *e.g.*, the trivial multi-view constraints that the model can only be supervised from sparse viewpoints, and the high interdependence between Gaussian attributes make their optimization ambiguity, *i.e.*, optimizing the position *vs* optimizing the shape. Although some recent efforts [23, 65] have attempted to use monocular depth priors [36, 37] to stabilize the optimization, *e.g.*, the monocular depth consistency of sampled virtual viewpoints [65] and the hard-soft monocular depth regularization [23], as shown in Fig. 1, the inherent scale and multi-view inconsistency of monocular depth make it hard to guarantee a consistent scene structure and lead to unsatisfactory rendering results, especially in complex scenes.

To this end, we are motivated to exploit the matching prior, which exhibits worthwhile characteristics indicating the ray/pixel correspondence between views and the multi-view visible region. Based on this, we propose *SCGaussian*, a framework that leverages matching priors to explicitly enforce the optimization of scene structure to be 3D consistent. A straightforward idea for this purpose is to use the ray correspondence to supervise the reprojection error of the rendering depth. However, we observe that the rendering geometry is not always consistent with the scene structure due to the interdependence of Gaussian attributes. In this paper, we argue that in addition to the rendering geometry, the more important aspect to ensure the consistency of the scene structure is to optimize the *position of Gaussian primitives*. To achieve this, we design a hybrid representation, which consists of ray-based Gaussian primitives besides the ordinary non-structure Gaussian primitives. For these rays-based ones, we bind them to matching rays and restrict the optimization of their position along the ray, thus we can utilize the matching correspondence to optimize the position of Gaussian primitives to converge to the consistent surface position where rays intersect. In this dual optimization solution, both the position and shape of the Gaussian primitives can be constrained properly.

Extensive experiments on LLFF [30], IBRNet [52], DTU [16], Tanks and Temples [21] and Blender [29] datasets show the effectiveness of our SCGaussian, which is capable of synthesizing detail and accurate novel views in these forward-facing, surrounding, and complex large scenes, achieving new state-of-the-art performance in both rendering quality (**3 − 5 dB** improvement on challenging complex scenes [21]) and efficiency (∼**200 FPS** rendering and 1-minute convergence speed).

## 2    Related Works

**Novel view synthesis.** Novel view synthesis is a task to render realistic images of unseen views given a set of training images. Many methods are proposed to address this problem in both traditional [7, 4, 45, 22, 55] and deep-learning based [64, 63, 11, 10] manners. In particular, NeRF [29] achieves photo-realistic rendering and has become one of the most popular methods in recent years, which

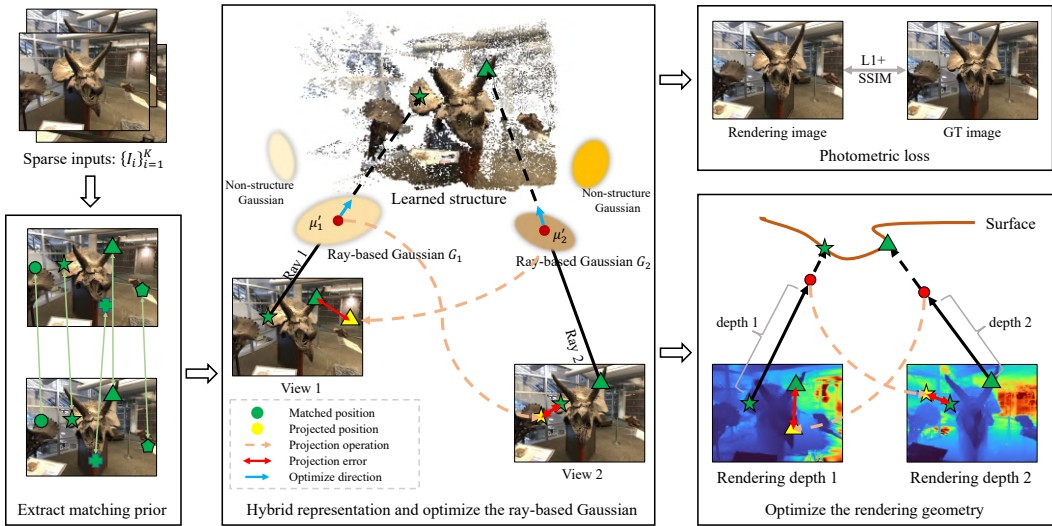

Figure 2: **Framework of SCGaussian.** We first extract the matching prior from the sparse input, and randomly initialize the hybrid Gaussian representation. The ray-based Gaussian primitives are bound to matching rays, and are explicitly optimized using the matching correspondence. The rendering geometry optimization is further conducted to optimize the shape of all types of Gaussian primitives. Combined with the ordinary photometric loss, SCGaussian can learn the consistent scene structure.

successfully combines multi-layer perceptrons (MLP) and volume rendering. The following works try to improve NeRF in many aspects, *e.g.*, quality [1, 2], pose-free [54, 28, 25, 50], dynamic view synthesis [35, 12, 24, 33], training and rendering efficiency [14, 31, 13, 47, 5, 26]. And more recently, a point-based method 3D Gaussian Splatting [19] represents the scene as 3D Gaussians and significantly improve the rendering speed to a real-time level. And it has shown an advantage in many aspects [56, 59, 61, 18, 27, 49, 3] compared with NeRF-based methods. However, these methods need dense input views, which makes them unsuitable for many practical applications.

**Few-shot novel view synthesis.** Compared with the ordinary NVS, the few-shot NVS is a more practical task but also more challenging. The original rendering methods always suffer from dramatic degradation when applied directly to the sparse scenario. Many works have attempted to solve this problem. Specifically, one thread of works [60, 52, 17, 6] attempt to pre-train a generalizable model on the large-scale datasets first and apply it to the target scene with sparse inputs. Another alternate approach is to optimize the model from scratch for each scene. [8, 39] try to add the depth supervision from the SFM points or depth completion model, and [62, 51] adopt the more practical monocular depth prior. To exploit smoothness and semantic priors, works [32, 15] choose to render some patches first and introduce the geometry and appearance regularization. These methods are all based on the NeRF and rely on volume rendering to synthesize novel views, which is always time-consuming. Some recent methods [65, 23, 57] combine the efficient 3DGS representation with monocular depth [36] and multi-view stereo [34] prior to improve the efficiency of the few-shot NVS task. However, since 3DGS relies on the initialization of sparse SFM points and adequate multi-view constraints, which are hard to observe in the sparse scenario, how to learn the globally consistent structure is the crucial bottleneck.

## 3  Methodology

In this section, we introduce the proposed new few-shot approach, SCGaussian, which can learn consistent 3D scene structure using matching priors. The overall framework of our model is illustrated in Fig. 2. In Sec. 3.1, we first review the 3DGS. Then we elaborate on the challenge of few-shot 3DGS and the motivation of using matching priors in Sec. 3.2, and the design of our Structure Consistent Gaussian Splatting will be introduced in Sec. 3.3. The full loss function and training detail will be described in Sec. 3.4.

## 3.1 Preliminary of 3D Gaussian Splatting

3DGS [19] explicitly represents the 3D scene through a collection of anisotropic 3D Gaussians. Each Gaussian is defined by a center vector $\mu \in \mathbb{R}^3$ and a covariance matrix $\Sigma \in \mathbb{R}^{3\times3}$, and the influence for a position $x$ is defined as:

$$G(x) = e^{-\frac{1}{2}(x-\mu)^T\Sigma^{-1}(x-\mu)}. \tag{1}$$

To ensure positive semi-definiteness and effective optimization, the covariance matrix is decomposed into a scaling matrix $S$ and a rotation matrix $R$ as:

$$\Sigma = RSS^T R^T, \tag{2}$$

where these two matrices can be derived by a scaling factor $s \in \mathbb{R}^3$ and a rotation factor $r \in \mathbb{R}^4$. Additionally, each Gaussian also contains the appearance feature $sh \in \mathbb{R}^k$ represented by a set of spherical harmonics (SH) coefficients and an opacity value $\alpha \in \mathbb{R}$.

To render the image, the 3DGS is projected to the 2D image plane via a view transformation matrix $W$ and the Jacobian of the affine approximation of the projective transformation $J$:

$$\Sigma' = JW\Sigma W^T J^T. \tag{3}$$

Using the point-based rendering, the color $C$ of a pixel can be computed by blending $N$ ordered Gaussians overlapping the pixel:

$$C = \sum_{i\in N} c_i \alpha'_i \prod_{j=1}^{i-1}(1-\alpha'_j), \tag{4}$$

where $c_i$ denotes the view-dependent color of $i$-th Gaussian, and $\alpha'_i$ is determined by the multiplication of $\Sigma'$ and the opacity $\alpha_i$. Similarly, we can also render the depth image $D$ through:

$$D = \sum_{i\in N} d_i \alpha'_i \prod_{j=1}^{i-1}(1-\alpha'_j), \tag{5}$$

where $d_i$ refers to the depth of $i$-th Gaussian.

In summary, each Gaussian point $\theta$ can be characterized by the set of attributes: $\theta = \{\mu, s, r, \alpha, sh\}$. To optimize the model, 3DGS takes the photometric loss, which is measured with the combination of L1 and SSIM [53], between the rendering image $\hat{I}$ and the ground-truth image $I$:

$$L_{photo} = (1-\lambda)L_1(\hat{I}, I) + \lambda L_{ssim}(\hat{I}, I), \tag{6}$$

where $\lambda$ is always set to 0.2. Facilitated by the highly-optimized rasterization pipeline, 3DGS can achieve remarkably fast rendering and training speed and enables real-time view synthesizing.

## 3.2 Motivation of Matching Priors

Through mimicking the image-formation process at training views, the model aims to find a set of optimal Gaussians $\mathcal{G} = \{\theta_i\}_{i=1}^N$ to build a photo-realistic mapping function $f_{\mathcal{G}} : P \rightarrow \hat{I}$, i.e., mapping the image for arbitrary camera pose $P$:

$$\mathcal{G} = \arg\min_{\mathcal{G}} L_{photo}(\mathcal{G}). \tag{7}$$

With adequate training views, the optimized model is capable of generating great novel view rendering results. However, as shown in Fig. 3, when the input becomes sparse, the 3DGS model always overfits training views and suffers from a significant degradation in test poses.

We observe that the challenge of the few-shot 3DGS mainly comes from the failure of learning the 3D consistent scene structure, e.g., the learned Gaussian primitives cannot distribute over the accurate surface region and the rendering geometry is multi-view inconsistent. In the sparse scenario, the supervision signal only comes from a few training poses, and this trivial multi-view constraint makes it hard to bias the model towards learning a 3D consistent solution. Conversely, as shown in Fig. 3 (a), 3DGS model tends to learn the inconsistent Gaussians for each view separately, e.g.,

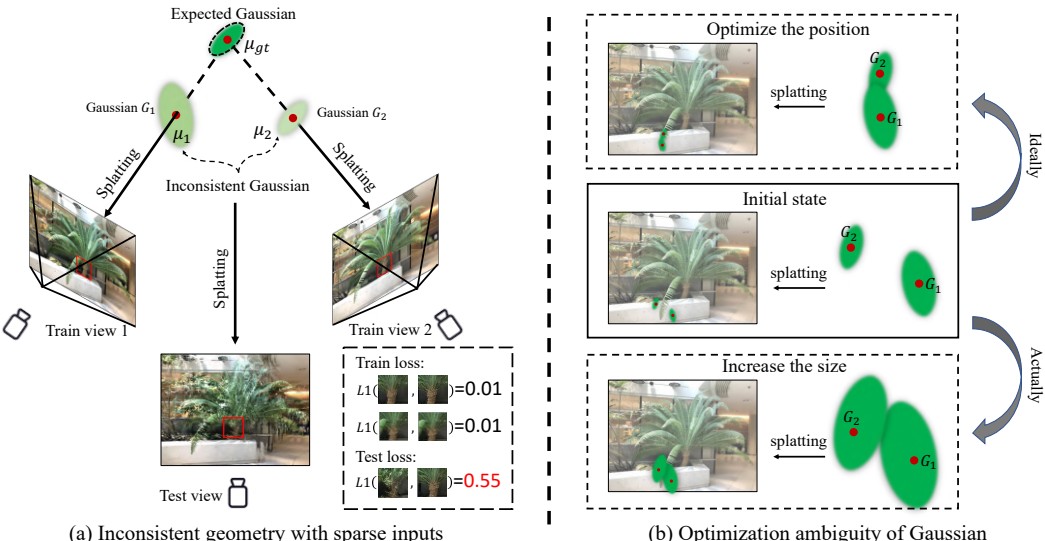

(a) Inconsistent geometry with sparse inputs      (b) Optimization ambiguity of Gaussian

Figure 3: **Visualization of some challenges faced by few-shot 3DGS.** (a) The expected Gaussian in the surface region cannot be learned, and the model tends to learn the inconsistent Gaussian and overfit the training views. While the training loss is small enough, the testing error is pretty bad. (b) The attributes of Gaussian primitives are interdependent and the model tends to increase the size to cover the pixels rather than correct the position.

the model learns a "wall" extremely close to each camera, in which case the training loss is still small. Furthermore, we find that 3DGS has an obvious optimization ambiguity due to the high interdependence of attributes, *e.g.*, shape versus position. Theoretically, the model needs to learn more small-sized Gaussian primitives over the texture region to recover high-frequency details, but in practice, it prefers to increase the size of Gaussian primitives to cover these pixels as shown in Fig. 3 (b), resulting in an overly smooth view synthesis. To ensure that the learned structure is consistent, a heuristic strategy in the vanilla 3DGS is to use sparse SFM points as initialization and guide the model's optimization, which is especially crucial for complex scenes. However, in the sparse scenario, it's pretty hard to stably extract enough SFM points, and usually, only random initialization can be used like [23]. This amplifies the challenge of learning the consistent structure.

Although some methods [65, 23] attempt to use the monocular depth to regularize the geometry, the inherent scale and multi-view ambiguity of monocular depth make it difficult to solve the aforementioned problems. Thus, we are interested in the question: *how can we make 3DGS without SFM point initialization to learn 3D consistent scene structure under sparse input?* In this paper, we consider exploiting matching priors using the pre-trained matching model [44], which doesn't face the ambiguity problem like monocular depth. Matching priors have two important characteristics: ray correspondence and ray position.

**Ray correspondence.** The pair of matching rays represent the corresponding 2D position of a consistent 3D point in different views, which can serve as the prominent multi-view constraint, *i.e.*, the matching rays should theoretically intersect at the same surface position. Given a pair of matching rays $\{r_i, r_j\}$ at image $I_i$ and $I_j$, and the corresponding pixel coordinates are $\{p_i, p_j\}$, supposing we have computed the position of the surface point intersect with each ray as $X_i$ and $X_j$, we can get the following equation:

$$X_i = X_j. \tag{8}$$

Meanwhile, given the camera intrinsics $\{K_i, K_j\}$ and extrinsics $\{[R_i, t_i], [R_j, t_j]\}$, we can further project the surface point to another 2D image plane and get the projected pixel coordinate, *e.g.*, the projection from $i$ to $j$ can be modeled as:

$$p_{i \to j}(X_i) = \pi(K_j R_j^T (X_i - t_j)), \tag{9}$$

where $\pi$ is the projection operator $\pi([x, y, z]^T) = [x/z, y/z]^T$. Thus we have the equation in pixel coordinate: $p_j = p_{i \to j}$, and similarly, we have $p_i = p_{j \to i}$.

**Ray position.** In the matching prior, the position of matching rays exactly indicates the region that is commonly visible to at least two views. This multi-view visible region plays a crucial role in the reconstruction model, as it's meaningless when there is no overlapping region between views. In the sparse scenario, the stereo correspondence is insignificant and the non-overlapping region can even harm the model training, while the importance of the multi-view visible region is magnified.

### 3.3 Structure Consistent Gaussian Splatting

To fully exploit the characteristics of matching prior, our SCGaussian explicitly optimizes the scene structure in two folds: the position of Gaussian primitive and the rendering geometry. Optimizing the position of Gaussian primitive is non-trivial due to the non-structural properties of Gaussian primitives. To address this, we present a hybrid Gaussian representation. Besides ordinary non-structure Gaussian primitives used to recover the background region visible in a single view, our model also consists of ray-based Gaussian primitives which are bound to matching rays, in which case their positions are restricted to be optimized along the ray.

**Initialization and densification.** Different from existing methods that initialize with either SFM points [65] or random points [23], we initialize with ray-based Gaussian primitives and bind them to matching rays. For convenience, here we discuss two input images $I_i$ and $I_j$. Suppose we have $N$ pairs of matching rays $\{r_i^k, r_j^k\}_{k=1}^N$, we can initialize $N$ pairs of ray-based Gaussian primitives $\{\mathcal{G}_i^k, \mathcal{G}_j^k\}_{k=1}^N$. Similar to 3DGS, each primitive is equipped with a set of learnable attributes but with a different position representation. The position of the ray-based Gaussian primitive $\mu'$ is defined as:

$$\mu' = o + zd, \tag{10}$$

where $o$ and $d$ refer to the camera center and ray direction respectively, and $z$ is a learnable distance factor, which is randomly initialized.

For densification, we follow the same strategy in [19] to determine the "under-reconstruction" candidates using the average magnitude of view-space position gradients, and generate the non-structure Gaussian primitives, whose positions can be optimized in arbitrary directions.

**Optimize the position of Gaussian primitives.** As analyzed in Sec. 3.2, the accurate position of Gaussian primitives plays a fundamental role in the learned scene structure. Since the matching correspondence between ray-based Gaussian primitives can be constructed using the binding strategy, we can conveniently optimize their positions.

For a pair of matching rays $\{r_i, r_j\}$ in image $I_i$ and $I_j$, thanks to our binding strategy, we can get a pair of binding Gaussian primitives $\{\mathcal{G}_i, \mathcal{G}_j\}$, and their positions in 3D space are $\mu'_i = o_i + z_i d_i$ and $\mu'_j = o_j + z_j d_j$ respectively. According to Eq. (8) and Eq. (9), we can get the projected 2D coordinate from $i$ to $j$: $p_{i \to j}(\mu'_i)$ and from $j$ to $i$: $p_{j \to i}(\mu'_j)$. Thus we can get the projection error of this pair of Gaussian primitives as:

$$\begin{cases} L_{gp}^{i \to j} = \|p_j - p_{i \to j}(\mu'_i)\| \\ L_{gp}^{j \to i} = \|p_i - p_{j \to i}(\mu'_j)\|. \end{cases} \tag{11}$$

The final Gaussian position loss $L_{gp}$ is computed as the average error of all binding Gaussian pairs.

**Optimize the rendering geometry.** Due to the interdependence of Gaussian attributes, the rendering geometry is not consistent with Gaussian positions, *e.g.*, the incorrect scaling or rotation can lead to wrong rendering geometry and affect the rendering results even with the correct Gaussian position.

We first render the depth image $D_i$ and $D_j$ through Eq. (5) and get the estimated depth for the pair of matching rays $\{D_i(p_i), D_j(p_j)\}$. Then we lift the pixel coordinate to 3D space:

$$\nu_i = R_i(D_i(p_i)K_i^{-1}\tilde{p}_i)) + t_i, \tag{12}$$

where $\tilde{p}$ refers to the 2D homogeneous of $p$. Similarly, we can get the 3D position $\nu_j$ of ray $j$. Then we get the projected 2D coordinate $p_{i \to j}(\nu'_i)$ and $p_{j \to i}(\nu'_j)$ according to Eq. (9) as mentioned above and compute the projection error based on the rendering depth as:

$$\begin{cases} L_{rg}^{i \to j} = \|p_j - p_{i \to j}(\nu'_i)\| \\ L_{rg}^{j \to i} = \|p_i - p_{j \to i}(\nu'_j)\|, \end{cases} \tag{13}$$

and we take the average error of all ray pairs as the final rendering geometry loss $L_{rg}$.

Table 1: **Quantitative comparisons on the LLFF and IBRNet datasets with 3 training views.** Best results are in **bold**. We run our method 5 times and report the error bar in the appendix.

| Method | Approach | LLFF | | | | IBRNet | | | |
|---|---|---|---|---|---|---|---|---|---|
| | | PSNR ↑ | SSIM ↑ | LPIPS ↓ | AVG ↓ | PSNR ↑ | SSIM ↑ | LPIPS ↓ | AVG ↓ |
| Mip-NeRF [1] | NeRF-based | 14.62 | 0.351 | 0.495 | 0.246 | 15.83 | 0.406 | 0.488 | 0.223 |
| RegNeRF [32] | | 19.08 | 0.587 | 0.336 | 0.149 | 19.05 | 0.542 | 0.377 | 0.152 |
| FreeNeRF [58] | | 19.63 | 0.612 | 0.308 | 0.134 | 19.76 | 0.588 | 0.333 | 0.135 |
| SparseNeRF [51] | | 19.86 | 0.624 | 0.328 | 0.127 | 19.90 | 0.593 | 0.364 | 0.137 |
| 3DGS [19] | 3DGS-based | 16.46 | 0.440 | 0.401 | 0.192 | 17.79 | 0.538 | 0.377 | 0.166 |
| FSGS [65] | | 20.43 | 0.682 | 0.248 | - | 19.84 | 0.648 | 0.306 | 0.130 |
| DNGaussian [23] | | 19.12 | 0.591 | 0.294 | 0.132 | 19.01 | 0.616 | 0.374 | 0.151 |
| **SCGaussian (Ours)** | | **20.77** | **0.705** | **0.218** | **0.105** | **21.59** | **0.731** | **0.233** | **0.097** |

## 3.4 Overall pipeline

**Loss function.** Our loss function consists of three parts: the ordinary photometric loss $L_{photo}$, the Gaussian position loss $L_{gp}$, the rendering geometry loss $L_{rg}$, and the full function is defined as:

$$L = L_{photo} + \beta L_{gp} + \delta L_{rg}. \tag{14}$$

**Training details.** During training, we set $\beta = 1.0$. To avoid the model falling into sub-optimization in the early stage of training, we set $\delta = 0$ and then increase it to $\delta = 0.3$ after 1k iterations. To ensure that the Gaussian primitive converges to the optimal position, we use a caching strategy in the first 1k iterations, *i.e.*, cache the position with the minimum Gaussian position loss $L_{gp}$ at each iteration. Meanwhile, considering there are some mismatched ray pairs in the matching prior, we further filter out those primitives with large Gaussian position loss $L_{gp} > \eta$. During optimization, the ray-based primitive will not be pruned. We build our model based on the official 3DGS codebase, and train the model for 3k iterations with the same setting as 3DGS but set the learning rate of the learnable distance factor $z$ to 0.1 at the beginning and decrease to $1.6 \times 10^{-6}$.

## 4 Experiments

In this section, we demonstrate the performance of our model in popular datasets and conduct ablation studies to verify the effectiveness of our designs. Next, we first describe the common datasets and the selected baselines for comparison, then analyze the results.

**Datasets & metrics.** We evaluate our model on forward-facing, complex large-scale and surrounding datasets under the sparse setting: LLFF [30], IBRNet [52], Tanks and Temples (T&T) [21], DTU [16] and NeRF Blender Synthetic dataset (Blender) [29]. LLFF dataset contains 8 real scenes, and following previous methods [32, 51], every 8-th images are held out for testing, and sparse views are evenly sampled from the remaining images for training. IBRNet dataset is also a real forward-facing dataset, and we select 9 scenes for evaluation and adopt the same split as in LLFF. T&T is a large-scale dataset collected from more complex realistic environments containing both indoor and outdoor scenes, and we use 8 scenes for evaluation and also apply the same split as in LLFF. DTU is an object-centric dataset, which contains more texture-poor scenes. We use the same evaluation strategy as [32] on DTU. For Blender, containing 8 object-centric synthetic scenes, we follow [58] to train with 8 images and test on 25 images. We report PSNR, SSIM, and LPIPS scores to measure our reconstruction quality and also report the geometric average (AVG) of $\text{MSE} = 10^{-\text{PSNR}/10}$, $\sqrt{1 - \text{SSIM}}$ and LPIPS as in [32].

**Baselines.** We compare our model against both NeRF-based and 3DGS-based few-shot NVS methods. For NeRF-based methods, we compare with methods with relatively high performance, including MipNeRF [1], DietNeRF [15], RegNeRF [32], FreeNeRF [58] and SparseNeRF [51]. For 3DGS-based methods, we compare with the vanilla 3DGS and its recent few-shot follow-ups like FSGS [65] and DNGaussian [23].

### 4.1 Results

**Results on LLFF and IBRNet.** We use the aforementioned split method to sample 3 images for training. The quantitative comparisons on two datasets with recent SOTA methods are summarized in Tab. 1. Although 3DGS-based methods natively have a weakness in invisible areas due to their

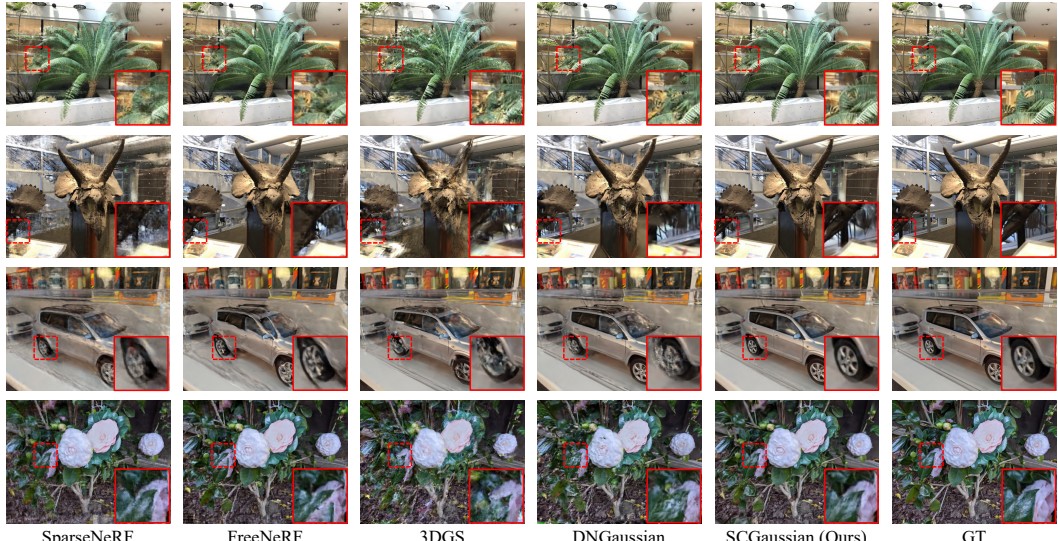

|            |           |      |            |                  |    |
| SparseNeRF | FreeNeRF | 3DGS | DNGaussian | SCGaussian (Ours) | GT |

Figure 4: **Qualitative comparisons on LLFF (first two rows) and IBRNet (last two rows) datasets with 3 training views.** The reconstruction of our method is more accurate and exhibits finer details.

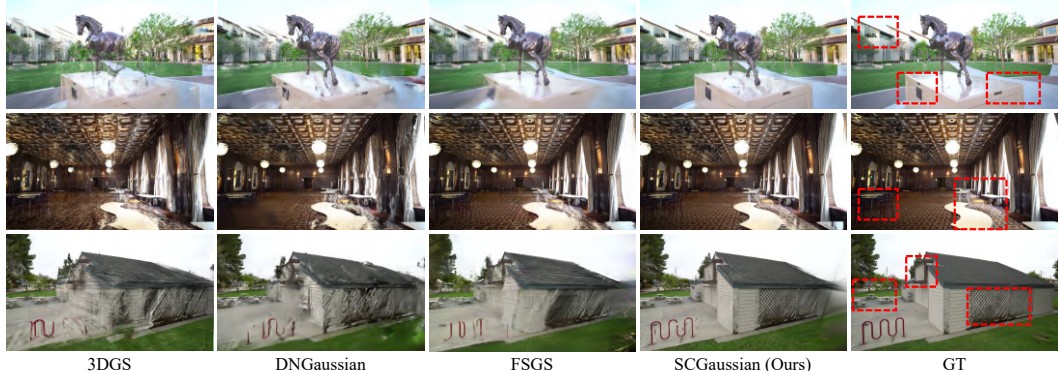

|      |            |      |                   |    |
| 3DGS | DNGaussian | FSGS | SCGaussian (Ours) | GT |

Figure 5: **Qualitative comparisons on Tanks and Temples dataset with 3 training views.**

discrete properties as discussed in [23], our SCGaussian still achieves the best performance in all metrics. Note that FSGS [65] uses the sparse SFM points for initialization, even though, our model holds remarkable superiority. Moreover, our advantage against previous methods is amplified in the IBRNet dataset, which has more low-texture scenes. Some qualitative comparisons are shown in Fig. 4, from which we can see that our method can recover more accurate high-frequency details.

**Results on T&T.** To evaluate the performance of our model on complex large scenes, we conduct further comparisons on the T&T dataset. Using the same split strategy as LLFF, we quantitatively compare with existing methods with 3 training views in Tab. 2. With the large difference in camera poses and the unbounded scene range, previous NeRF-based methods [51, 58, 32, 1], mostly designed for the bounded scenes, are hard to reconstruct plausible results. Among them, methods [32, 51] using geometric regularization perform better. Even combined with explicit point representation, the recent 3DGS still struggles on this large scene with sparse inputs. Although some recent efforts apply the monocular depth prior [23] or the initialization of sparse SFM point [65] to 3DGS, they have limited reconstruction quality. From the qualitative comparisons shown in Fig. 5, we can see that our method can synthesize the novel view with more accurate and complete details. Benefiting from our novel design in

Table 2: **Quantitative comparisons on the T&T dataset with 3 training views.**

| Method | PSNR ↑ | SSIM ↑ | LPIPS ↓ |
|---|---|---|---|
| MipNeRF [1] | 12.57 | 0.241 | 0.623 |
| RegNeRF [32] | 13.12 | 0.268 | 0.618 |
| FreeNeRF [58] | 12.30 | 0.308 | 0.636 |
| SparseNeRF [51] | 13.66 | 0.331 | 0.615 |
| 3DGS [19] | 17.14 | 0.493 | 0.397 |
| FSGS [65] | 20.01 | 0.652 | 0.323 |
| DNGaussian [23] | 18.59 | 0.573 | 0.437 |
| **SCGaussian (Ours)** | **22.17** | **0.752** | **0.257** |

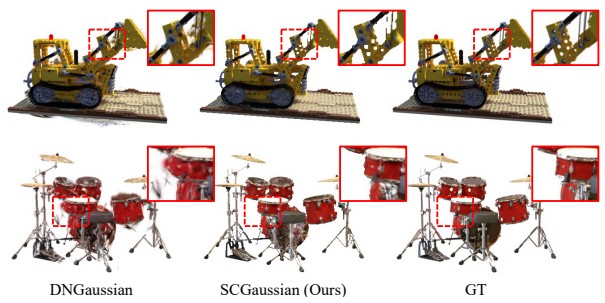

DNGaussian     SCGaussian (Ours)     GT

Figure 6: **Qualitative comparisons on Blender dataset.**

Table 3: **Quantitative comparisons on the Blender dataset.**

| Method | PSNR ↑ | SSIM ↑ | LPIPS ↓ |
|---|---|---|---|
| NeRF [29] | 14.934 | 0.687 | 0.318 |
| NeRF Simple [15] | 20.092 | 0.822 | 0.179 |
| Mip-NeRF [1] | 20.890 | 0.830 | 0.168 |
| DietNeRF [15] | 23.147 | 0.866 | 0.109 |
| DietNeRF + ft [15] | 23.591 | 0.874 | 0.097 |
| FreeNeRF [58] | 24.259 | 0.883 | 0.098 |
| SparseNeRF [51] | 22.410 | 0.861 | 0.119 |
| 3DGS [19] | 22.226 | 0.858 | 0.114 |
| FSGS [65] | 24.640 | **0.895** | 0.095 |
| DNGaussian [23] | 24.305 | 0.886 | 0.088 |
| **SCGaussian (Ours)** | **25.618** | 0.894 | **0.086** |

Table 4: **Quantitative comparisons on the DTU dataset with 3 training views.**

| Method | PSNR ↑ | SSIM ↑ | LPIPS ↓ | AVG ↓ |
|---|---|---|---|---|
| FreeNeRF | 19.92 | 0.787 | 0.182 | 0.098 |
| SparseNeRF | 19.55 | 0.769 | 0.201 | 0.102 |
| DNGaussian | 18.91 | 0.790 | 0.176 | 0.102 |
| Ours | **20.56** | **0.864** | **0.122** | **0.078** |

Table 5: **Comparisons with methods of directly initializing with triangulation points,** on LLFF.

| Method | PSNR ↑ | SSIM ↑ | LPIPS ↓ | AVG ↓ |
|---|---|---|---|---|
| 3DGS (baseline) | 16.46 | 0.440 | 0.401 | 0.192 |
| Triang. init + 3DGS | 19.11 | 0.643 | 0.335 | 0.140 |
| Triang. init + ScaffoldGS | 19.41 | 0.699 | 0.217 | 0.118 |
| Triang. init + OctreeGS | 19.61 | 0.710 | 0.210 | 0.114 |
| Ours | **20.77** | **0.705** | **0.218** | **0.105** |

hybrid representation and explicit optimization of rendering geometry and position of Gaussian primitives, our model can learn more consistent structure as shown in Fig. 1, and show great generalization ability on these large scenes.

**Results on DTU.** We further conduct more experiments on DTU dataset to prove the robustness on more texture-poor scenes. The quantitative results in Tab. 4 indicate that our method achieves the best performance on all metrics, which proves that our model is still robust on those texture-poor scenes. The qualitative results in Fig. 7 also demonstrate that our method can recover more accurate details.

**Results on Blender.** We test on the Blender dataset to verify our performance in the surrounding scenario. While [65] uses its uppooling strategy to clone more Gaussian primitives and gets the best SSIM score, our method achieves the best PSNR and LPIPS scores, as the quantitative results reported in Tab. 3. We visualize more qualitative comparisons in Fig. 6 to demonstrate our superiority, and we can see that our method has a clear advantage in recovering fine details and reconstructing complete structures. This further demonstrates our generalization ability in different scenes.

## 4.2 Analysis

**Ablation studies.** We conduct a few ablation studies on LLFF and T&T datasets to understand how our model performs with different settings. Our baseline is the vanilla 3DGS. From the results shown in Tab. 6, we can see that using only the hybrid representation (Hybrid Rep.), our model can already bring more than 2dB improvement to the baseline, which verifies that our model can indeed mitigate the risk of overfitting. Combined with the optimization of rendering geometry (Rend. Geo.), the performance can be further improved. When we optimize both the rendering geometry and the position of Gaussian primitives (Dual optim.), the model can learn the more consistent scene structure and render more convincing novel views. These results prove our motivation for learning the 3D consistent structure. And the adopted cache & filter strategy further mitigates the impact of wrong matching priors. To assess the performance of the model with different numbers of views, we conduct the comparison in T&T dataset, as shown in Fig. 8. Our model can

Table 6: **Ablation studies on LLFF dataset.**

| Method | PSNR ↑ | SSIM ↑ | LPIPS ↓ |
|---|---|---|---|
| Baseline | 16.46 | 0.440 | 0.401 |
| w/ Hybrid Rep. | 18.62 | 0.607 | 0.273 |
| w/ Rend. Geo. | 19.40 | 0.634 | 0.259 |
| w/ Dual optim. | 20.69 | 0.703 | **0.205** |
| w/ Cache & filter | **20.77** | **0.705** | 0.218 |

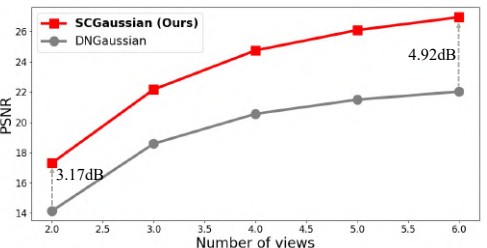

Figure 8: **PSNR *vs* view number on T&T.**

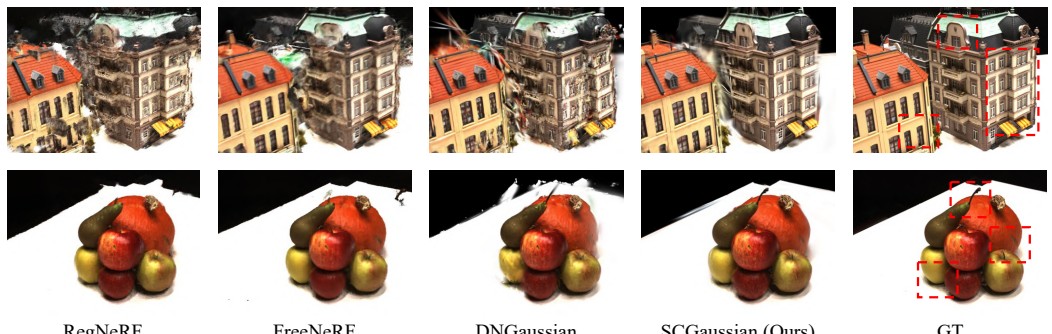

| RegNeRF | FreeNeRF | DNGaussian | SCGaussian (Ours) | GT |

Figure 7: **Qualitative comparisons on DTU dataset with 3 training views.**

consistently outperform the SOTA method [23], and the advantage becomes more significant as the number of views increases.

**Triangulation initialization.** To prove the effectiveness of our optimization strategy, we perform some comparisons with methods that directly use the triangulation points of matched pixels for initialization. The results are shown in Tab. 5, which indicate that using the triangulation initialization can improve the performance of the baseline especially equipped with more structured ScaffoldGS [27] or OctreeGS [38] models. Even though, our model still achieves the best performance and demonstrates the effectiveness of our model.

**Robustness to matching models.** We perform more experiments in Tab. 7 to verify the robustness of our model to different pre-trained matching models. Concretely, we use the same optimization and testing configuration for all models, and additionally use the DKM [9], LoFTR [48] and SuperGlue [40] models to extract the matching prior. The results in Tab. 7 show that

Table 7: **Robustness to different matching models,** conducted on the LLFF dataset.

| Method | PSNR ↑ | SSIM ↑ | LPIPS ↓ | AVG ↓ |
|---|---|---|---|---|
| Ours + GIM | 20.77 | 0.705 | 0.218 | 0.105 |
| Ours + DKM | 20.92 | 0.732 | 0.189 | 0.099 |
| Ours + LoFTR | **20.94** | **0.737** | **0.182** | **0.097** |
| Ours + SuperGlue | 20.25 | 0.689 | 0.221 | 0.110 |

all these matching models can bring a satisfactory improvement to the baseline, and our method can even achieve better performance when using weaker matching models (*e.g.*, GIM *vs.* LoFTR). These results prove that our strategy is robust to different matching models.

**Efficiency.** With a single NVIDIA RTX 3090 GPU, the training of our method consumes about 3GB memories and converges within 1 minute on LLFF 3-view setting, which is much faster than existing methods, *e.g.*, [58, 51] need about 10 hours and [65] needs about 10 minutes. Our method also achieves a real-time inference speed of over 200FPS at $504 \times 378$ resolution, superior to NeRF-based methods (*e.g.*, [58] at 0.04FPS) and comparable to 3DGS-based methods (*e.g.*, [23] at 181FPS).

**Limitation.** Following the common pipeline in the research field of few-shot NVS, our model requires an accurate camera pose, which may not always be available. Thus liberating this limitation could further improve our work to be more practical, and we will investigate this in future work.

## 5 Conclusion

In this paper, we observed the main challenge of few-shot 3DGS is learning the 3D consistent scene structure, and we exploited the matching prior to construct a Structure Consistent Gaussian Splatting method named *SCGaussian*. Due to the optimization ambiguity of Gaussian attributes between the position and shape, we presented two approaches to optimize the scene structure: explicitly optimize the rendering geometry and the position of Gaussian primitives. While directly constraining the position is non-trivial in the vanilla 3DGS, we introduced a hybrid Gaussian representation, consisting of ordinary non-structure Gaussian primitives and ray-based Gaussian primitives. In this way, both the position and shape of Gaussian primitives can be optimized to be 3D consistent. To evaluate our method as comprehensively as possible, we conducted experiments on forward-facing, complex large-scale, and surrounding datasets. The results consistently demonstrate that our method achieves new state-of-the-art performance while being highly efficient.

## Acknowledgments and Disclosure of Funding

This work is financially supported by the Outstanding Talents Training Fund in Shenzhen, this work is also supported by the National Natural Science Foundation of China U21B2012, Shenzhen Science and Technology Program-Shenzhen Cultivation of Excellent Scientific and Technological Innovation Talents project(Grant No. RCJC20200714114435057). J. Jiao is supported by the Royal Society Short Industry Fellowship (SIF\R1\231009) and the Amazon Research Award. In addition, we sincerely thank all assigned anonymous reviewers, whose comments were constructive and very helpful to our writing and experiments.

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

# A   Appendix

## A.1   More experimental details

Our experiments are performed following the common solution of existing methods. LLFF dataset contains eight different scenes, and we perform the training and inference at $8\times$ downsampling scale with a resolution of $504 \times 378$. IBRNet is another forward-facing dataset collected by [52] that contains larger camera motion and more scenes. We select nine scenes for evaluation, which include 'giraffe_plush', 'yamaha_piano', 'sony_camera', 'Japanese_camilia', 'scaled_model', 'dumb-bell_jumprope', 'hat_on_fur', 'roses' and 'plush_toys'. We use the same setting as LLFF. Tanks and Temples is a large-scale complex dataset, which has large camera motion. For comparison, we use eight scenes, namely 'Ballroom', 'Barn', 'Church', 'Family', 'Francis', 'Horse', 'Ignatius', and 'Museum', both indoors and outdoors. We train and infer at the resolution of $960 \times 540$. For the Blender dataset, we use the common solution in existing methods and run at the resolution of $400 \times 400$ ($2\times$ downsampling).

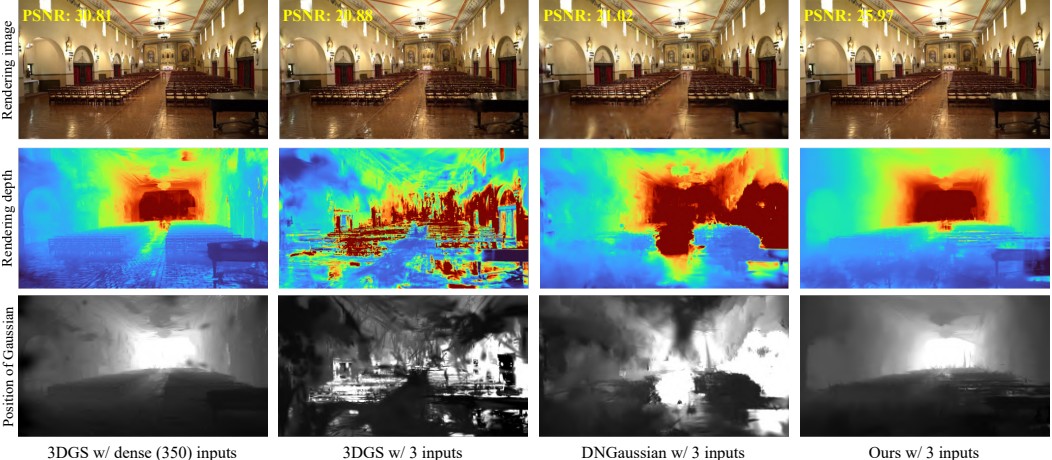

Figure 9: **Comparisons in view synthesis, rendering geometry and position of Gaussian primitive.** The first, second and third rows show the synthesized novel view of four methods, their rendering depth and the position of nearest Gaussian primitives, respectively.

## A.2   More results of consistent structures

The consistent scene structure is important for reconstruction models, including the NeRF-based and the 3DGS-based, and the inaccurate structure, *e.g.*, floaters or walls, can lead to extremely poor novel view synthesis results. Thus in this paper, we propose SCGaussian to solve the challenge of learning consistent structure in few-shot 3DGS models. From the quantitative and qualitative results shown in our main paper, we can see that our method can synthesize more complete novel views, especially in the high-frequency regions, and these results just demonstrate our learned scene structure is more 3D consistent.

We show some comparisons of rendering geometry in Fig. 1, and we can see that when the input becomes sparse, existing methods fail to learn the plausible geometry while our method can still render the accurate geometry. Here, we show more comparisons in Fig. 9 to understand the results of the position of Gaussian primitives. To visualize these positions, we first fix the opacity of all primitives to a large value (1.0 in our setting) and then we render the distance of Gaussian primitives using the Gaussian rasterization. In this way, the rendering results can indicate the position of the nearest Gaussian primitives (third row in Fig. 9). We can see that both the rendering geometry and the position of Gaussian primitives of our method are more accurate and 3D consistent.

Meanwhile, we find that our learned structure in texture-less regions (*e.g.*, the wall region shown in Fig. 9 3rd row) is even better (smoother) than the dense version (with way more views of inputs) of 3DGS. We suspect the main reason is that the proposed method has better control over the number of Gaussian primitives, *i.e.*, adaptively allocates more primitives in high-textured regions while fewer primitives in the texture-less regions.

## A.3 Effectiveness of dual optimization based on the hybrid representation

In this paper, we propose a dual optimization strategy to separately optimize the rendering geometry and position of Gaussian primitives based on our hybrid Gaussian representation. While we show some quantitative results in Tab. 6, here we show more visual results in Fig. 10. The model 'w/ Matching prior' in Tab. 6 refers to the straightforward combination of matching priors and the vanilla 3DGS, and the model 'w/ Dual optim' corresponds to the model optimizes both rendering geometry and position of Gaussian primitives based on the hybrid representation.

We can see that the straightforward solution is still hard to synthesize accurate novel views and still suffers from obvious inconsistencies in its rendering geometry. And our solution, optimizing both the rendering geometry and position of Gaussian primitives based on our hybrid representation, can synthesize more accurate novel views and render more consistent depth.

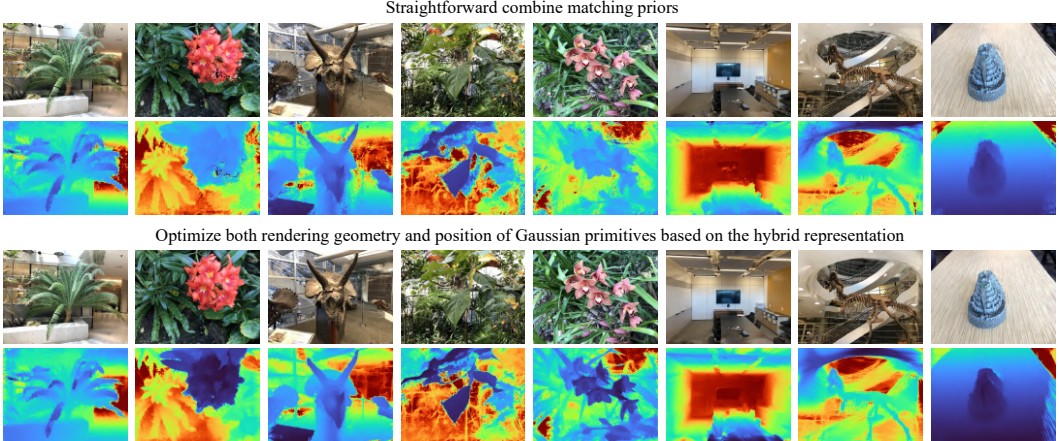

Figure 10: **Results of using the straightforward combination and the dual optimization.**

## A.4 Results at different resolutions

While 3DGS model can synthesize high-resolution images efficiently, we here conduct more comparisons with existing methods at a higher resolution ($1008 \times 756$) on LLFF dataset. As the quantitative results shown in Tab. 8, our method still achieves the best in all metrics. Compared with the previous few-shot 3DGS method [23], our advantage is amplified at the higher resolution.

We further show some visual comparisons in Fig. 11. We can see that our method can recover more high-frequency details with the best accuracy, while previous 3DGS-based method [23] even loses some structures. Compared with the NeRF-based methods [58, 51], which have a slow rendering speed and smooth reconstruction, our advantage is more significant.

Table 8: **Quantitative comparisons on LLFF under different resolutions.** Experiments are conducted with 3 training views.

| Method | res 8× (504 × 378) | | | res 4× (1008 × 756) | | |
|---|---|---|---|---|---|---|
| | PSNR ↑ | SSIM ↑ | LPIPS ↓ | PSNR ↑ | SSIM ↑ | LPIPS ↓ |
| MipNeRF [1] | 14.62 | 0.351 | 0.495 | 15.53 | 0.416 | 0.490 |
| RegNeRF [32] | 19.08 | 0.587 | 0.336 | 18.40 | 0.545 | 0.405 |
| FreeNeRF [58] | 19.63 | 0.612 | 0.308 | 19.12 | 0.568 | 0.393 |
| SparseNeRF [51] | 19.86 | 0.620 | 0.329 | 19.30 | 0.565 | 0.413 |
| 3DGS [19] | 16.46 | 0.401 | 0.440 | 15.92 | 0.504 | 0.370 |
| DNGaussian [23] | 19.12 | 0.591 | 0.294 | 18.03 | 0.574 | 0.394 |
| **SCGaussian (Ours)** | **20.77** | **0.705** | **0.218** | **20.09** | **0.679** | **0.252** |

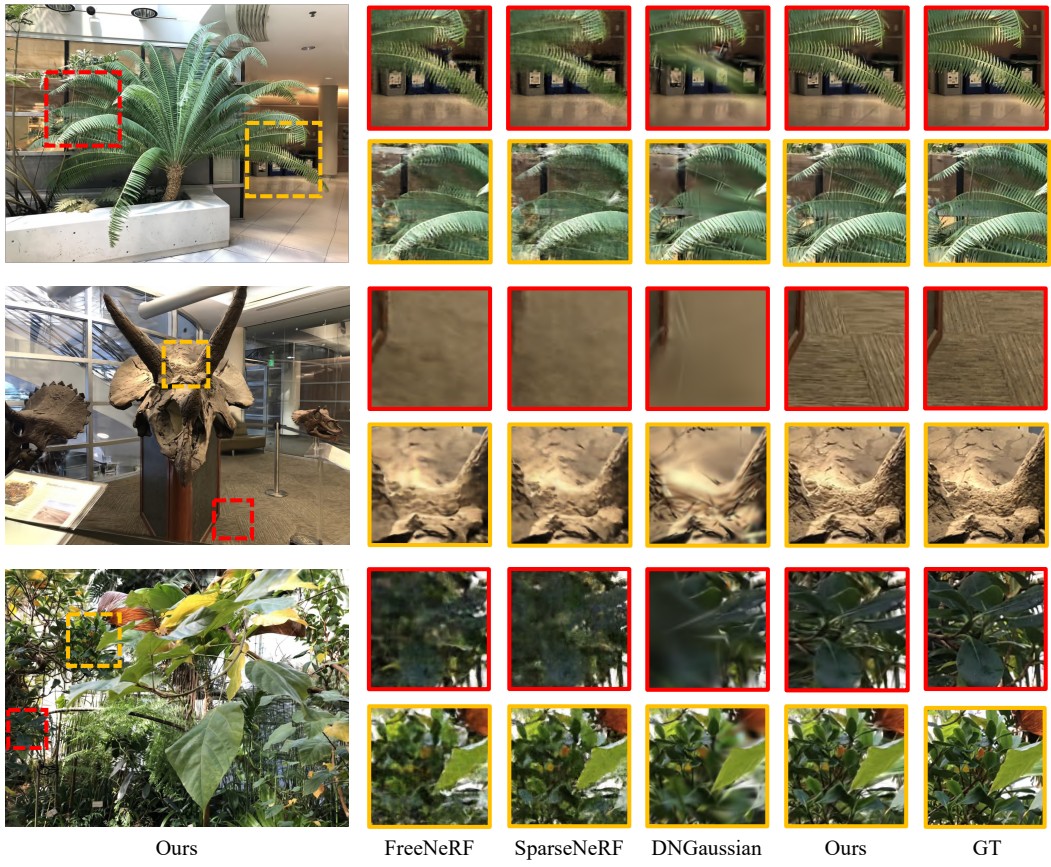

| Ours | | FreeNeRF | SparseNeRF | DNGaussian | Ours | GT |

Figure 11: **Qualitative comparisons at the high resolution** ($1008 \times 756$).

## A.5 Results for different view numbers

As shown in Fig. 8, our method can consistently outperform existing methods with different numbers of inputs. Here, we show some visual comparisons in Fig. 12 to qualitatively evaluate our advantage. We can see that our method can recover more details with both 3 and 6 training views. Furthermore, we report some quantitative comparisons in Tab. 9. The results show that the 3DGS-based methods perform better than the NeRF-based method on the complex scene. FreeNeRF [58] propose efficient frequency regularization terms to improve the few-shot performance in the bounded scene, but we can see that this simple strategy does not work well in the unbound scene, while SparseNeRF [51] uses the monocular depth prior to achieve better performance than FreeNeRF. This suggests that using external priors may be a better option in complex scenarios. Meanwhile, we find a situation that the vanilla 3DGS performs better than DNGaussian, which adopts the monocular depth to regularize the geometry. We analyze that the main reason is the inherent scale and multi-view inconsistency of monocular depth. This situation further demonstrates the superiority of the matching prior that we adopt.

Table 9: **Quantitative comparisons on T&T with different numbers of inputs.**

| Method | 3 views | | | 6 views | | |
| | PSNR ↑ | SSIM ↑ | LPIPS ↓ | PSNR ↑ | SSIM ↑ | LPIPS ↓ |
|---|---|---|---|---|---|---|
| FreeNeRF [58] | 12.30 | 0.308 | 0.636 | 14.34 | 0.375 | 0.586 |
| SparseneRF [51] | 13.66 | 0.331 | 0.615 | 17.50 | 0.454 | 0.539 |
| 3DGS [19] | 17.14 | 0.493 | 0.397 | 22.27 | 0.702 | 0.275 |
| DNGaussian [23] | 18.59 | 0.573 | 0.437 | 22.03 | 0.687 | 0.382 |
| **SCGaussian (Ours)** | **22.17** | **0.752** | **0.257** | **26.95** | **0.869** | **0.149** |

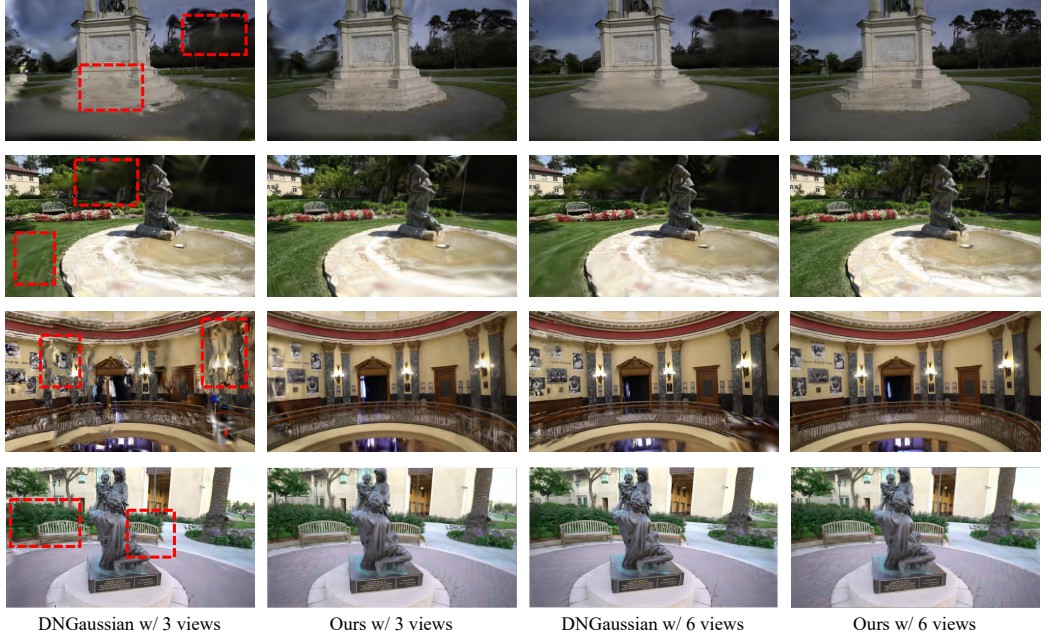

| DNGaussian w/ 3 views | Ours w/ 3 views | DNGaussian w/ 6 views | Ours w/ 6 views |

Figure 12: **Qualitative comparisons with 3 and 6 training views.**

## A.6    More discussion on the hybrid representation

Optimizing the position of Gaussian primitives to the 3D consistent surface position is fundamental for the novel view synthesis task. However the Gaussian primitive in the vanilla 3DGS is non-structure and is hard to be controlled, whose position can be moved to arbitrary directions. While the dense counterpart can leverage the initialization of SFM points to guide the optimization of the Gaussian primitive, our few-shot model with sparse inputs can only start from the random initialization, which obviously makes the optimization of the position of Gaussian primitives become more difficult. Thus, it would be an ideal solution if there was a method that could directly control the position of the Gaussian primitive.

With the ray correspondence in the matching prior, we can assume that there is a surface point in the matching ray. Therefore, we propose to bind Gaussian primitives to matching rays, restrict the optimization of their positions along the ray and enforce them to converge to the surface position. This approach makes the optimization of Gaussian primitives more controllable. Meanwhile, we notice that there are still regions that are not multi-view visible, only using these ray-based Gaussian primitives makes it hard to cover the complete scene, as shown in Fig. 13. Here, we treat these regions only visible to a single view as the 'background', and we use the ordinary non-structure Gaussian primitives to recover them and propose the hybrid representation. We report the ablation results of the hybrid representation in Tab. 8.

Table 10: **Ablation results of the hybrid representation on LLFF with 3 training views.**

| Method | PSNR ↑ | SSIM ↑ | LPIPS ↓ |
|---|---|---|---|
| only non-structure | 19.40 | 0.634 | 0.259 |
| only ray-based | 20.50 | 0.684 | 0.231 |
| hybrid rep. | **20.77** | **0.705** | **0.218** |

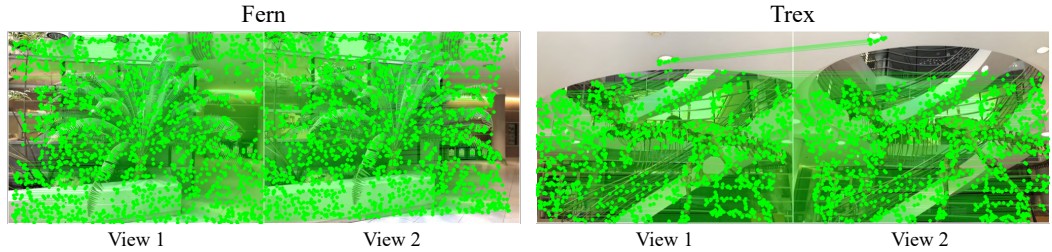

| Fern | Trex |
| View 1    View 2 | View 1    View 2 |

Figure 13: **Visualization of matching points on two scenes.**

### A.7  Error bars

Although most previous don't provide the error bar, here, to enhance the experimental significance, we run all methods 5 times and report error bars of SparseNeRF [51], 3DGS [19], DNGaussian [23] and our method in Fig. 14. We can see that the results of the baseline model 3DGS [19] have the largest fluctuations in all metrics. Our method gets the best score on all metrics and has relatively satisfactory stability.

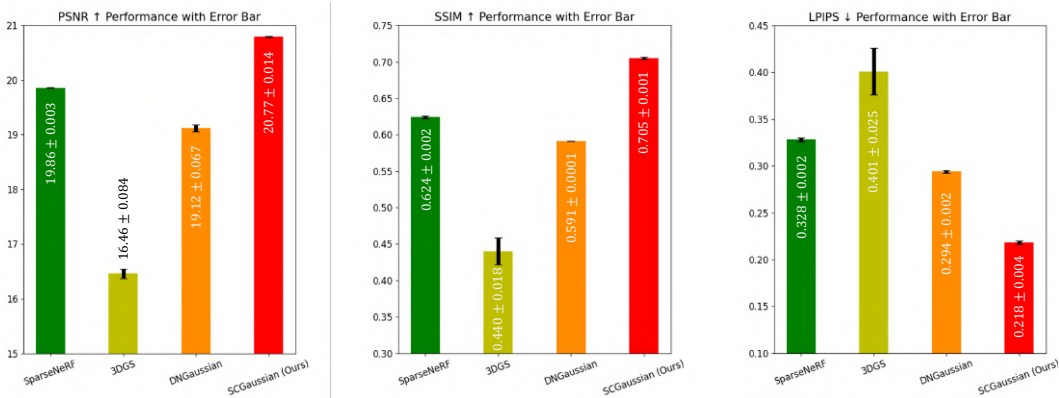

Figure 14: **Error bars of SparseNeRF [51], 3DGS [19], DNGaussian [23] and our method.**

### A.8  Video comparisons

To better illustrate the effectiveness of our method, here we further provide visual comparisons in video format. Please refer to https://drive.google.com/drive/folders/1sTpuRRV4YYJOPTpQYb-ZChrck37GtU6h?usp=sharing for more details.

