# OpenReview forum: "Structure Consistent Gaussian Splatting with Matching Prior for Few-shot Novel View Synthesis"
_NeurIPS.cc/2024/Conference — NeurIPS 2024 poster_

### Official Review · Reviewer_PwZQ · 2024-07-10

**Soundness:** 3
**Presentation:** 3
**Contribution:** 3
**Rating:** 5
**Confidence:** 4

**Summary:**

The paper proposes SCGaussian, a few-shot 3D Gaussian Splatting model to address novel view degeneration in sparse input scenarios. SCGaussian leverages matching priors to enforce 3D consistency by optimizing the position of Gaussian primitives along rays, overcoming challenges in monocular depth-based methods. This hybrid representation binds ray-based and non-structure Gaussian primitives, ensuring accurate scene structure. Extensive experiments demonstrate SCGaussian's state-of-the-art performance, achieving significant improvements in rendering quality and efficiency.

**Strengths:**

1. The paper is written clearly, making it easy to follow.
  2. The experiment is very comprehensive.
  3. The proposed method achieves state-of-the-art performance in the task of few-shot NVS, despite not requiring sparse 3D points as input like other works did.
  4. The combination of proposed optimization of the position of Gaussian primitives (section 3.3) between  optimization of the rendering geometry is quite novel. The ablation studies of section 4.2 clearly demonstrate the impact of dual optimization.

**Weaknesses:**

- The paper claims it does not require SFM points for initialization. However, it heavily relies on precomputed camera poses. When using COLMAP to obtain camera poses, SFM points are generated effortlessly as a byproduct. Alternatively, if we have camera poses and 2D correspondences, why not use triangulation to obtain 3D points?

  - Again, I am curious about the advantages of the proposed optimization of Gaussian primitives' positions (Section 3.3) over triangulation. Given that SVD in triangulation is significantly easier compared to optimizing a learnable depth parameter, it would be more convincing to include experiments evaluating NVS quality, generalization, optimization time, and computational resources.

  - The novelty of the paper is somewhat limited and may not meet the high standards expected at NIPS.

  - The author mentioned sparse input in large scenes, and I'm curious if the proposed method can be generalized to even larger scenes, such as driving scenes like Waymo or aerial photography data. For instance, in driving scenes, there is often a challenge of losing matching to the road area due to its lack of texture.

  - The details regarding the matching prior should be more comprehensive, including information such as the average number of pairs extracted, the time it takes for the extraction process, the hyperparameters used in the model, and other relevant details.

  - The evaluation would be more convincing with the inclusion of additional geometric metrics, such as Chamfer Distance.

**Questions:**

All questions are present in weakness.

**Limitations:**

Limitations have been disscussed in the paper.

---

> ### Author Rebuttal · Authors · 2024-08-07
>
> We sincerely thank reviewer #42ua for recognizing our work and the valuable comments. Here, we will address the concerns point by point.
>
> **Q: It heavily relies on precomputed camera poses. When using COLMAP to obtain camera poses, SFM points are generated effortlessly as a byproduct.**
>
> - As the common setting of the few-shot NVS field, our target scene is the applications which have off-the-shuffle camera poses but only have sparse cameras, e.g., the poses between the sparse cameras in a driving car is known from the sensor or pre-computed (Not from COLMAP).
> - In these sparse scenarios, the traditional COLMAP method is hard to extract SFM points, and the SFM points in existing public datasets (e.g., LLFF dataset) are generated from dense views. Therefore, for a fair comparison with NeRF methods, we don't use these SFM points for initialization.
>
> **Q: Comparisons with the direct triangulation.**
>
> As suggested, we further conducted comparisons with methods that directly initialize existing methods with the triangulation point clouds, **in Tab. 3 and Fig. 2 of our uploaded PDF**. Besides the 3DGS, we also test on the more advanced methods like ScaffoldGS [2] and OctreeGS [3].
> - The results indicate that initializing with the triangulation points indeed improve the performance of existing methods. However, this improvement only comes from the better initialization and this strategy cannot mitigate the impact of the wrong matching priors. And the initialized Gaussian primitives face the same non-structure problem like the vanilla 3DGS.
> - Our model can achieve more powerful performance because of our further designs on hybird representation and dual optimization, whose effectiveness has been demonstrated **in Tab. 1 of our uploaded PDF**.
> - For a fair comparison, we train these models for more iterations. On a single RTX 3090 GPU, the optimization of the model based on 3DGS, ScaffoldGS and OctreeGS takes about 5m, 10m, and 13m, respectively. Their inference speed is about 220FPS, 160FPS and 135FPS, respectively. And as declared in the Efficiency section (from Line 292 to Line 296), our model runs at about 200FPS, converges in 1 minutes and still achieves the best performance.
>
> **Q: More explanation of our novelty.**
>
> - To our knowledge, we are *the first method* to attempt to integrate matching priors to the few-shot 3DGS and successfully mitigate the degradation in sparse scenarios.
> - We propose a hybrid Gaussian representation consisting of ray-based Gaussian and the vanilla non-structure Gaussian, which restrics the Gaussian primitive move on the ray, making the Gaussian primitive more controllable and simplifing the model convergence space. This simplification is beneficial for few-shot NVS tasks, since the complex model tends to overfit, as demonstrated by DietNeRF [4] and FreeNeRF [5].
> - We bind the ray-based Gaussian primitive to the matched ray and enable the direct optimization of the position of the Gaussian primitive besides the optimization of the rendering geometry. Combined with the ordinary photometric loss, we can properly alleviate the optimization ambiguity between the position and size of Gaussian primitives in the few-shot scenario.
> - The ablation studies **in Table 1 of our uploaded PDF** demonstrate the effectiveness of each contribution.
>
> **Q: More experiments on Waymo dataset.**
>
> As suggested, we further conducted experiments on larger scenes. But since the full Waymo dataset is too large, we conducted comparisons with DNGaussian [6] on the sample scene provided by UC-NeRF [1], due to the time limit of rebuttal. Specifically, we took the middle 20 frames for experiments and trained the model with 5 input views at the resolution of 4x downsampling.
>
> The quantitative comparisons are shown in the table below. Our method has a clear performance gain over the previous method DNGaussian. We show some visualization results **in Fig. 3 of our uploaded PDF**, from which we can see that our method can recover more details, including the texture of the road.
>
> | Method      | PSNR$\uparrow$ | SSIM$\uparrow$ | LPIPS$\downarrow$ | AVG$\downarrow$ |
> | :--- | :----:   | :----: | :----: | :----: |
> | DNGaussian (CVPR2024) | 20.46 | 0.736 | 0.326 | 0.116 |
> | SCGaussian (Ours) | **28.72** | **0.906** | **0.156** | **0.043** |
>
> **Q: More details of the matching prior.**
>
> We use the GIM model in our paper, and we use the *default hyperparameters* of GIM. In the setting of three training views, the matching model can extract about *5k pairs* per image pairs and takes about *5 seconds* per scene.
>
> **Q: The inclusion of geometric metrics, such as Chamfer Distance.**
>
> Thanks for this suggestion, and we have tried to conduct such evaluation on the DTU dataset. The results show that our $Thre_{10}$ metric (Percentage of pixels with error more than 10mm) of rendered depth is about 30% and NDGaussian is more than 99%. We speculate that there are some errors and mismatches between the camera pose and the ground-true depth. We will try to conduct the experiment on other suitable synthetic datasets.
>
> And as the common practice of existing few-shot methods, we reported the qualitative comparisons of the rendering depth in Fig. 1 of our main paper, where our method performs much better in geometric accuracy, showing its superiority in geometry.
>
> **Reference**
>
> [1] UC-NeRF: Neural Radiance Field for Under-Calibrated Multi-view Cameras in Autonomous Driving, ICLR2024.
>
> [2] Scaffold-GS: Structured 3D Gaussians for View-Adaptive Rendering, CVPR2024.
>
> [3] Octree-GS: Towards Consistent Real-time Rendering with LOD-Structured 3D Gaussians, arxiv2024.
>
> [4] Putting NeRF on a Diet: Semantically Consistent Few-Shot View Synthesis, ICCV2021.
>
> [5] FreeNeRF: Improving Few-Shot Neural Rendering With Free Frequency Regularization, CVPR2023.
>
> [6] DNGaussian: Optimizing Sparse-View 3D Gaussian Radiance Fields with Global-Local Depth Normalization, CVPR2024.

---

### Official Review · Reviewer_42ua · 2024-07-10

**Soundness:** 3
**Presentation:** 3
**Contribution:** 3
**Rating:** 7
**Confidence:** 5

**Summary:**

The paper proposes a structure guided novel view synthesis method for Gaussian Splatting. Experiments show that the proposed method produces clearer rendering results than other existing works.

**Strengths:**

* The proposed method is well evaluated with various methods in neural rendering. Experiments is sufficient to support the claims.
* Demonstrations show that the proposed method is simple but efficient.

**Weaknesses:**

* The proposed method employ a pre-trained matching model to obtain matching priors. It is a strong dependency. A discussion / ablation study on the effect of its performance is necessary.

* In addition, failure cases under any special scene should be discussed in the limitation. For example, the surrounding views are not always available in trafic scenes so that there are serious occlusions/dynamics in observation. How about the performance under such condition?

**Questions:**

Please refer to the weaknesses.

**Limitations:**

Discussion over failure cases should be included.

---

> ### Author Rebuttal · Authors · 2024-08-07
>
> We sincerely thank the reviewer #42ua for recognizing our work and the valuable comments. As suggested, we conduct more ablation studies with different types of matching models and add more discussion of the failure cases.
>
> **Q: A discussion / ablation study on the pre-trained matching model (matching priors).**
>
> To study the effect of the matching priors to our model, **in Tab. 2 of our uploaded PDF**, besides the GIM prior (ICLR2024) [1] we used in our paper, we further conducted experiments with three more matching models, including SuperGlue (CVPR2020) [2], LoFTR (CVPR2021) [3] and DKM (CVPR2023) [4].
> - The quality of the matching prior can indeed affect the performance of the model, but the results show that all these four kinds of matching priors can bring satisfactory improvement to the baseline. And this proved our designs are robust to different matching models.
> - It's worth noting that our model has the cache and filter strategy, which can mitigate the negative effect of the wrong matching prior. Even in the worst case where there is no exact matching prior, our model can still achieve better performance than the baseline model, when using our hybrid representation, which has been proved in the results of **Tab. 1 of our uploaded PDF** ("+ Hybrid Rep." vs. "3DGS (Baseline)").
>
> **Q: More discussion of the failure cases.**
>
> We appreciate the great suggestion! We summarize some failure cases as follows:
> - Our method is a reconstruction model, which requires the multi-view coverage. Therefore, it struggles to reconstruct the accurate scene structure when there is *severe* occlusion between views. And for this kind of scene, leveraging priors in generative models may be a feasible solution.
> - The texture-less background is also a challenging problem for our model. This is an ill-posed problem, and we found that the implicit NeRF methods are much better in these regions because of their good smoothness. To solve this, we argue that more effective geometric supervision or a more generalizable model trained in large datasets is necessary.
>
> **Reference**
>
> [1] GIM: Learning Generalizable Image Matcher From Internet Videos, ICLR2024.
>
> [2] SuperGlue: Learning feature matching with graph neural networks, CVPR2020.
>
> [3] Loftr: Detector-free local feature matching with transformers, CVPR2021.
>
> [4] Dkm: Dense kernelized feature matching for geometry estimation, CVPR2023.

---

> > ### Comment · Reviewer_42ua · 2024-08-12
> >
> > Very interesting results. I will keep my score.

---

### Official Review · Reviewer_nvzF · 2024-07-10

**Soundness:** 3
**Presentation:** 3
**Contribution:** 3
**Rating:** 5
**Confidence:** 4

**Summary:**

This work presents a Structure Consistent 3DGS method using matching priors to learn 3D consistency. A hybrid Gaussian representation, including non-structure Gaussian primitives and ray-based Gaussian primitives, is introduced. Position Consistency loss between two views is adopted to achieve multi-view alignment. Extensive experiments prove the effectiveness of the proposed method.

**Strengths:**

1. This paper is well-written and easy to understand.
2. The concept of the ray-based Gaussian primitives is interesting, which can solve the issue that 3DGS tends to increase the size of Gaussian primitives.
3. The proposed method achieves SOTA performance. The ablation study has also verified the effectiveness of the proposed method.

**Weaknesses:**

1. The optimization of the rendering geometry in lines 195-203 is not novel, which is similar to the correspondence pixel reprojection loss in CorresNeRF [A].
2. Which network is used for feature matching?
3. For the ray-based Gaussian primitives, do they have the shape? If not, can we call them Gaussian primitives?
3. How do the authors select the value of N? Is it defined by the results of feature matching?
4. Why does this work in line 177 define N pairs of ray-based Gaussian primitives rather than N ray-based Gaussian primitives? Since The matched pixels should have the same 3D surface points.

[A] Lao, Yixing, et al. "Corresnerf: Image correspondence priors for neural radiance fields." Advances in Neural Information Processing Systems 36 (2023): 40504-40520.

**Questions:**

Please refer to the Weaknesses.

**Limitations:**

The authors have clarified the limitations.

---

> ### Author Rebuttal · Authors · 2024-08-07
>
> We sincerely thank reviewer #nvzF for your time and valuable comments. We noticed that the main concerns are the differences with CorresNeRF and more explanation of the implementation details. Besides, we will address all concerns point by point.
>
> **Q1: The differences with CorresNeRF.**
>
> First of all, thanks for pointing out this important related work, and we will cite and add more analysis in our final version. Here, we give some analysis as follows:
>
> One of our main designs is the *dual optimization*, instead of solely optimizing the rendering depth like CorresNeRF. We argue that optimizing the position of the Gaussian primitives is more important than optimizing the rendering geometry for learning consistent structures (Lines 51-56), because the rendering geometry is not consistent with the scene structure due to the interdependence of Gaussian attributes, a problem that NeRF methods do not face. The results **in Tab. 1 of our uploaded PDF** illustrated this (Improvement of *1.37dB* of "Dual Optim." compared to "Rend. Geo.").
>
> **Q2: Which network is used for feature matching?**
>
> We adopt the GIM model for feature matching as declared in Line 147, and we further conducted with more different matching models **in Tab. 2 of our uploaded PDF**.
>
> **Q3: For the ray-based Gaussian primitives, do they have the shape? If not, can we call them Gaussian primitives?**
>
> Yes, the ray-based Gaussian primitives also have shape but with different representation of position，and thus we try to optimize both the position and shape of Gaussian primitives properly.
>
> **Q4: How do the authors select the value of N? Is it defined by the results of feature matching?**
>
> Yes, the value of N is determined by the results of feature matching.
>
> **Q5: Why does this work in line 177 define N pairs of ray-based Gaussian primitives rather than N ray-based Gaussian primitives?**
>
> We assign a pair of Gaussian primitives for each pair of matched rays, and this is more robust when the matching prior is not perfect. To demonstrate this, we conducted the comparison experiment shown in the table below.
>
> | Method      | PSNR$\uparrow$ | SSIM$\uparrow$ | LPIPS$\downarrow$ | AVG$\downarrow$ |
> | :--- | :----:   | :----: | :----: | :----: |
> | Single | 20.66 | 0.699 | 0.224 | 0.110 |
> | Pair (Ours) | **20.77** | **0.705** | **0218** | **0.105** |

---

> > ### Comment · Reviewer_nvzF · 2024-08-11
> > **Reply to the authors**
> >
> > Thank the authors for addressing my concerns. They have addressed my concerns, and I will raise my score to a borderline accept.

---

> > > ### Author Response · Authors · 2024-08-12
> > >
> > > We thank the reviewer for taking the time to review our response. We’re pleased that our rebuttal has addressed the concerns and we sincerely appreciate the reviewer's score upgrade. We will incorporate the key points the reviewer raised as we work on the revised paper.

---

### Official Review · Reviewer_k97m · 2024-07-11

**Soundness:** 2
**Presentation:** 2
**Contribution:** 3
**Rating:** 7
**Confidence:** 5

**Summary:**

This paper proposes SCGaussian that utilizes a pre-trained image matcher to get the dense pixel-wise matching correspondences between sparse observing views, then regard these as the prior for 3DGS to achieve high-quality few-shot novel view synthesis. It innovatively binds Gaussians to the matched pixels and restricts their position to be along the ray, which can keep the scene geometry close to that of the image matcher predicted. Experiments on various datasets demonstrated its superior performance and efficiency compared to existing methods.

**Strengths:**

- The ray binding is novel and interesting, which constrains the reconstructed scene geometry not too far from the matching correspondence, and thus avoids an overall structure collapse caused by overfitting.

- The division of structured and non-structured Gaussians is well-motivated.

- Experimental results are good in visualization and quantitative results.

**Weaknesses:**

- As the correspondences and camera poses are all already known, why do not just initialize the Gaussians just at the intersection point? Furthermore, the introduction of a more powerful image matcher conflicts with the motivation declared in lines 145-146, as once the high-quality matching correspondences are sufficient, SfM methods can also produce high-quality point clouds. The main motivation of this paper is not convincing.

- Despite the authors have conducted some experiments and tried to verify the effect of the proposed dual optimization strategy, it's unclear what is exact the setting of "w/ Matching prior" in Table 4 and "Straightforward combine matching priors" in Figure 9. Therefore, they are not convincing for me to verify the effect of the proposed strategy.

- The method seems to heavily rely on the image matcher, however, nearly all evaluation datasets have strong textures, which are friendly to image matching methods, and only one extremely powerful matcher GIM [39] is used for the method. Would like to see more evaluations to prove the method's robustness when the matching correspondence is not ideal, e.g. using more kinds of matcher, and reconstructing scenes with large texture-poor regions.


[39] Xuelun Shen, Zhipeng Cai, Wei Yin, Matthias Müller, Zijun Li, Kaixuan Wang, Xiaozhi Chen, and Cheng Wang. Gim: Learning generalizable image matcher from internet videos. In ICLR, 2024.

**Questions:**

- The main concern is that I'm not sure whether the improved performance is just from a more powerful matching prior or the proposed designs. If most regions can be correctly matched according to GIM or other matchers, it would be very easy to build a high-quality point cloud and take it as the initialization, which can also bring a good structure prior, especially when applying to some anchor-based works like Scaffold-GS [1]. Please refute this with experiments or reasoning to prove the necessity of the proposed designs.

- Would like to see more explanations for the second point of Weaknesses.

- To evaluate the robustness, experiments using more kinds of matcher, and reconstructing scenes with more texture-poor regions (like DTU) are expected to be added.


[1] Lu, Tao, et al. "Scaffold-gs: Structured 3d gaussians for view-adaptive rendering." Proceedings of the IEEE/CVF Conference on Computer Vision and Pattern Recognition. 2024.

**Limitations:**

The authors discussed the limitation of accurate camera pose requirement. Besides, this work is built upon one specified off-the-shelf image matcher, yet does not verify its robustness for other choices.

---

> ### Author Rebuttal · Authors · 2024-08-07
>
> We sincerely thank reviewer #k97m for recognizing our work and the valuable comments. We noticed that the main concerns are the performance of our designs and the robustness of different matching models and texture-poor scenes. Here, we will explain the specific questions individually:
>
> **Q: The performance and necessity of our designs. (Including the explanation of the second point of Weaknesses)**
>
> Our contribution is to exploit the matching prior to construct a structure consistent Gaussian splatting method. While leveraging matching prior to solve the challenge of few-shot 3DGS is one of our important contributions, our other main designs are in two folds:
> - we propose a hybrid Gaussian representation consisting of ray-based Gaussian and the vanilla non-structure Gaussian (Hybrid Rep.), which makes the Gaussian primitive more controllable and simplifies the model convergence space through restricting Gaussian primitives to only move on the ray.
> - we bind the ray-based Gaussian primitive to the matched ray and enable the direct optimization of the position of Gaussian primitive besides the optimization of the rendering geometry (Dual Optim.). Combined with the ordinary photometric loss, we can properly alleviate the optimization ambiguity between the position and size of Gaussian primitives in the few-shot scenario.
>
> To prove these, as suggested, we conducted more detailed ablation studies and comparisons **in Tab. 1, Tab. 3 and Fig. 2 of our uploaded one-page PDF**.
> - From Tab. 1, we can see that even without any matching priors, our "Hybrid Rep." can achieve more than 2dB improvement (from 16.46dB to 18.62dB), demonstrating the effectiveness of this design.
> - To validate the effectiveness of our "Dual optim." design, we compare with the model only constraining the rendering geometry (Rend. Geo.) in Tab. 1. And the "Rend. Geo." is just the same as the setting of "w/ Matching prior" and "Straightforward combine matching priors", and we are sorry for not unifying and explaining it (**Our explanation of the second point of Weaknesses**). The results prove that our "Dual optim." can more completely combine matching priors to achieve the best performance (2.15dB improvement over the "Hybrid Rep." and 1.37dB improvement over "Rend. Geo.").
> - As suggested, we further perform the comparison with methods initialize 3DGS, ScaffoldGS [1], and OctreeGS [2] models using point clouds directly triangulated from matching rays and camera poses, as shown **in Tab. 3 and Fig. 2 of our uploaded PDF**. Although they can achieve better performance than the vanilla 3DGS with better initialization, they struggle to mitigate the impact of the wrong matching samples and struggle to further optimize the position of Gaussian primitive like the vanilla 3DGS. On the contrary, our designs can efficiently optimize the position during the training, and our model has simpler convergence space with our hybrid representation, which is more suitable for few-shot NVS task and has been proved in DietNeRF [6] and FreeNeRF [7]. After all, since there may be surface points on this ray, there is no need to let the Gaussian primitives move throughout the entire 3D space.
>
> **Q: The robustness to different matching models.**
>
> As suggested, **in Tab. 2 of our uploaded PDF**, we conducted more experiments with different matching models, i.e., SuperGlue (CVPR2020) [3], LoFTR (CVPR2021) [4] and DKM (CVPR2023) [5].
> - The results indicate that the matching prior from all matching models can bring  satisfactory improvements to the baseline.
> - The quality of matching prior does have an impact on the quality of the reconstruction, but our model has the cache and filter strategy which can mitigate the impact of wrong matching to some extent as demonstrated in Tab. 4 of our paper.
>
> **Q: The robustness to more texture-poor regions.**
>
> As suggested, **in Tab. 4 and Fig. 1 of our uploaded PDF**, we performed the comparisons with existing methods on the more texture-poor DTU dataset.
> - The results show that our method can still achieve the best performance, especially compared to recent 3DGS method DNGaussian (CVPR2024) [8].
> - The low texture is an ill-posed and common problem faced by all methods, so the scene that cannot get the ideal matching correspondence is still difficult for existing methods. Our model can still recover a more consistent structure with the insufficient matching prior, with our hybrid representation and dual optimization designs.
>
> **Reference**
>
> [1] Scaffold-GS: Structured 3D Gaussians for View-Adaptive Rendering, CVPR2024.
>
> [2] Octree-GS: Towards Consistent Real-time Rendering with LOD-Structured 3D Gaussians, arxiv2024.
>
> [3] SuperGlue: Learning feature matching with graph neural networks, CVPR2020.
>
> [4] Loftr: Detector-free local feature matching with transformers, CVPR2021.
>
> [5] Dkm: Dense kernelized feature matching for geometry estimation, CVPR2023.
>
> [6] Putting NeRF on a Diet: Semantically Consistent Few-Shot View Synthesis, ICCV2021.
>
> [7] FreeNeRF: Improving Few-Shot Neural Rendering With Free Frequency Regularization, CVPR2023.
>
> [8] DNGaussian: Optimizing Sparse-View 3D Gaussian Radiance Fields with Global-Local Depth Normalization, CVPR2024.

---

> ### Comment · Reviewer_k97m · 2024-08-08
>
> Thanks for the authors' detailed reply. Most of my concerns are solved.
>
> Here I have some additional problems:
>
> 1) Despite SCGaussian outperforms Scaffold-GS and Octree-GS, there seems no special design to delete wrong matching pixels compared to Scaffold-GS and Octree-GS besides "Cache & filter", as the ray-based Gaussians can not be pruned in my understanding. Meanwhile, the method can still work well without "Cache & filter" according to Table 4. So, how SCGaussian escape from the "struggle to mitigate the impact of the wrong matching samples"? If the flexibility of ray-based Gaussians along the ray is the reason, wish this feature to be added to the paper to help understand the design of ray-based Gaussian.
>
> 2) As illustrated in Figure 3, the goal of eq.(11, 13) is to optimize a pair of Gaussians converge to the same surface position with the same appearance. Then, is it really necessary to initialize two ray-based Gaussians for one surface point? Only one ray-based Gaussian seems to be sufficient for the framework. The only change may be on eq.(11), which can be modified as the loss between Gaussian position and ray intersection point, and all other parts can still work well. Although there is an ablation study in the rebuttal for Reviewer nvzF, would like to see more explanation about why it work.
>
> 3) After reading the experiment and discussion in the rebuttal, it's obviously easier for me to understand the method's effect than just reading the manuscript. Hope these valuable parts can be added to the paper if possible.
>
> I'll temporarily keep my rating.

---

> > ### Author Response · Authors · 2024-08-09
> >
> > We sincerely appreciate the discussions and suggestions. Below are our new responses to these points:
> >
> > 1. Yes, as the reviewer said, the model without "Cache & filter" can still works well and outperform ScaffoldGS and OctreeGS and we suspect the reason behind is the proposed hybrid representation in the few-shot scenario (simplification of the model convergence space as we declared), and meanwhile, our strategy can optimize the position of Gaussian primitive along the ray direction during the full training process, which can get more accurate position and is more effective compared to the direct triangulation. We will revise our paper according to the suggestion in the final version.
> >
> > 2. First of all, we agree with the reviewer that initializing only one Gaussian primitive for a matching pair is sufficient for most scenes, but we  found initializing two Gaussian primitives for a matching pair works better as shown in the table of our response to Reviewer nvzF. The reasons that we think mainly come from three aspects:
> > - this strategy can initialize more number of Gaussian primitives even the two Gaussians correspond to one surface point, and this is beneficial to recover more high-frequency details. This aligns with the strategy of vanilla 3DGS, whose densification operation clones more Gaussians to the same position as the original Gaussian.
> > - this strategy is beneficial to recover the view-dependent effect. For the single Gaussian strategy, the single Gaussian needs to encode the radiance information from 360°, which is non-trivial especially with the sparse training views. And for the proposed two Gaussians strategy, we use two Gaussians to encode the radiance information, which can be regarded as the interpolation of these two Gaussians that have different shape and correspond to different views, and this makes the encoding of radiance information more effective.
> > - this strategy may be more robust to the wrong matches, because we can initialize more ray-based Gaussian points to achieve better performance.
> >
> > 3. We appreciate that the reviewer find the additional experiments and discussion in the rebuttal help the understanding. We will incorporate these parts into the final version of the paper.

---

> ### Comment · Reviewer_k97m · 2024-08-11
>
> - I do think it's because the ray-based Gaussians can flexibly move when the matching prior is not so accurate, while will not go to some strange positions that may easily lead to overfitting, due to the ray and matching constraints. This can overcome some bad initialization points while keeping enough constraints for simplification. It's reasonable, but I can't directly tell this point from the words in the paper. To avoid confusion with previous structure-consistent GSs, maybe the authors can add some straight discussions like this in the revision.
>
> - Well, I still think initializing two ray Gaussians is not so concise and elegant, personally. Nevertheless, it may not be a problem as no extra negative influence is introduced. The working principle is valuable to be further analyzed. The provided ablation study may be added to the paper.
>
> Thanks for the response. I have no further questions. May consider raising the rating if there are no new problems found by other reviewers. Looking forward to the open access to the code and model.

---

> > ### Author Response · Authors · 2024-08-12
> >
> > We sincerely thank the reviewer for the further valuable suggestions.
> > - Thanks for this suggestion, we'll add more analysis about the ray-based Gaussian and incorporate these discussions into our revised final version.
> > - As suggested, we will add the ablation study to the paper. Meanwhile, we will carefully consider a further analysis of the working principle.
> > - We will also release the code and models to be publicly available.

---

### Author Rebuttal · Authors · 2024-08-07

We sincerely thank all reviewers for providing constructive feedback that helped us improve the paper. We are encouraged that reviewers appreciated the methodology, experiments, and writing of our paper, and acknowledged that:
- the proposed method is novel, interesting, well-motivated and efficient (Reviewer #k97m, #nvzF, #42ua, #PwZQ).
- the experimental analysis is comprehensive and the results are good and SOTA (Reviewer #k97m, #nvzF, #42ua, #PwZQ).
- the writing is clear and easy to follow and understand (Reviewer #nvzF, #PwZQ).

We noticed that the main concerns of reviewers are concentrated on the request for more discussion of the robustness of different matching models and texture-poor scenes, more ablation study on the performance and necessity of our designs, and more explanation of the experiments and implementation details.

During the rebuttal phase, we have been working diligently on improving the paper on these fronts, addressing all the above concerns. Below, we briefly summarize the changes that we have made:
- we have performed more detailed ablation studies on our designs **in Tab. 1 of our uploaded PDF**. We mainly added a "Hybrid Rep." experiment which doesn't use any matching prior and with random initialization. The results show that our hybrid representation can bring more than *2dB* improvement than 3DGS. This is reasonable because our ray-based Gaussian restricts the optimization in the ray direction and simplifies the complexity, which is beneficial for few-shot NVS tasks, such as the small version of NeRF that has been demonstrated in DietNeRF [7] to achieve better performance than the original NeRF.
- we have conducted more experiments with different matching models (GIM (ICLR2024) [1], DKM (CVPR2023) [2], LoFTR (CVPR2021) [3] and SuperGlue (CVPR2020) [4]) **in Tab. 2 of our uploaded PDF**. The matching model can indeed affect the final performance, but the results show that our strategy can stably achieve SOTA performance with different matching models.
- we have compared with methods initialize existing vanilla 3DGS, ScaffoldGS [5], and OctreeGS [6] models using point clouds directly triangulated from matching rays and camera poses **in Tab. 3 and Fig. 2 of our uploaded PDF**. The results show that this simple strategy can indeed improve the performance especially combined with the anchor-based works ScaffoldGS and OctreeGS. However, these methods make it hard to mitigate the impact of the wrong matching samples and struggle to further optimize the position of Gaussian primitive like the vanilla 3DGS. Thus, our solution is more suitable for the few-shot NVS task and achieves better performance.
- we have further evaluated the texture-poor DTU dataset and the larger Waymo dataset, and the results are shown **in Tab. 4, Fig. 1 and Fig. 3 of our uploaded PDF**. In the texture-poor dataset, existing methods, even SOTA DNGaussian [8], struggle to reconstruct good results. Although the quality of the matching prior becomes worse, our method can filter out the wrong matching and achieve the best performance and reconstruct finest details, showing its robustness.

**Please refer to the response to each reviewer for more details and our uploaded one-page PDF for detailed experimental results**

**Reference**

[1] GIM: Learning Generalizable Image Matcher From Internet Videos, ICLR2024.

[2] Dkm: Dense kernelized feature matching for geometry estimation, CVPR2023.

[3] Loftr: Detector-free local feature matching with transformers, CVPR2021.

[4] SuperGlue: Learning feature matching with graph neural networks, CVPR2020.

[5] Scaffold-GS: Structured 3D Gaussians for View-Adaptive Rendering, CVPR2024.

[6] Octree-GS: Towards Consistent Real-time Rendering with LOD-Structured 3D Gaussians, arxiv2024.

[7] Putting NeRF on a Diet: Semantically Consistent Few-Shot View Synthesis, ICCV2021.

[8] DNGaussian: Optimizing Sparse-View 3D Gaussian Radiance Fields with Global-Local Depth Normalization, CVPR2024.

---

### Decision · Program_Chairs · 2024-09-25

**Decision:**

Accept (poster)

**Comment:**

The paper introduces a few-shot 3DGS method using a hybrid Gaussian approach, incorporating both ray-based and non-structured Gaussians. Starting with sparse views, a standard image matcher is utilized to compute the matching prior. Gaussians are then calculated for the matched pixels, ensuring their positions align with the rays, thereby preserving the scene geometry according to the matching prediction. Initial feedback from the reviewers is positive, highlighting significant strengths while also identifying issues that needed clarification. Despite some remaining concerns—such as (1) sensitive to various matching models, (2) challenges in texture-poor scenes, and (3) issues with strong occlusion—the rebuttal successfully addressed all other raised issues. As a result, the final ratings are 2 x A and 2 x BA. Reviewers have provided valuable suggestions for improving the paper, which the authors should consider during revisions.